# Automated Design using Neural Networks and Gradient Descent

## Abstract

We propose a novel method that makes use of deep neural networks and gradient decent to perform automated design on complex real world engineering tasks. Our approach works by training a neural network to mimic the fitness function of a design optimization task and then, using the differential nature of the neural network, perform gradient decent to maximize the fitness. We demonstrate this methods effectiveness by designing an optimized heat sink and both 2D and 3D airfoils that maximize the lift drag ratio under steady state flow conditions. We highlight that our method has two distinct benefits over other automated design approaches. First, evaluating the neural networks prediction of fitness can be orders of magnitude faster then simulating the system of interest. Second, using gradient decent allows the design space to be searched much more efficiently then other gradient free methods. These two strengths work together to overcome some of the current shortcomings of automated design.

## 1 Introduction

Automated Design is the process by which an object is designed by a computer to meet or maximize some measurable objective. This is typically performed by modeling the system and then exploring the space of designs to maximize some desired property whether that be an automotive car styling with low drag or power and cost efficient magnetic bearings (Ando et al., 2010) (Dyck & Lowther, 1996) . A notable historic example of this is the 2006 NASA ST5 spacecraft antenna designed by an evolutionary algorithm to create the best radiation pattern (Hornby et al.). More recently, an extremely compact broadband on-chip wavelength demultiplexer was design to split electromagnetic waves with different frequencies (Piggott et al., 2015). While there have been some significant successes in this field the dream of true automated is still far from realized. The main challenges present are heavy computational requirements for accurately modeling the physical system under investigation and often exponentially large search spaces. These two problems negatively complement each other making the computation requirements intractable for even simple problems.

Our approach works to solve the current problems of automated design in two ways. First, we learn a computationally efficient representation of the physical system on a neural network. This trained network can be used to evaluate the quality or fitness of the design several orders of magnitude faster. Second, we use the differentiable nature of the trained network to get a gradient on the parameter space when performing optimization. This allows significantly more efficient optimization requiring far fewer iterations then other gradient free methods such as genetic algorithms or simulated annealing. These two strengths of our method overcome the present difficulties with automated design and greatly accelerate optimization.

The first problem tackled in this work is designing a simple heat sink to maximize the cooling of a heat source. The setup of our simulation is meant to mimic the conditions seen with an aluminum heat sink on a computer processor. We keep this optimization problem relatively simple and use this only as a first test and introduction to the method. Our second test is on the significantly more difficult task of designing both 2D and 3D airfoils with high lift drag ratios under steady state flow conditions. This problem is of tremendous importance in many engineering areas such as aeronautical, aerospace and automotive engineering. Because this is a particularly challenging problem and often times unintuitive for designers, there has been considerable work using automated design to produce optimized designs. We center much of the discussion in this paper around this

problem because of its difficulty and view this as a true test our method. While we only look at these two problems in this work, we emphasize that the ideas behind our method are applicable to a wide variety of automated design problems and present the method with this in mind.

As we will go into more detail in later sections, in order to perform our airfoil optimization we need a network that predicts the steady state flow from an objects geometry. This problem has previously been tackled in Guo et al. (2016) where they use a relatively simple network architecture. We found that better perform could be obtained using some of the modern network architecture developments and so, in addition to presenting our novel method of design optimization, we also present this superior network for predicting steady state fluid flow with a neural network.

## 2 RELATED WORK

Because this work is somewhat multidisciplinary, we give background information on the different areas. In particular, we provide a brief discussion of other work related to emulating physics simulations with neural networks as this is of key importance in our method. We also review some of the prior work in automated design of airfoils because this is the main problem used to test our method.

## 3 SPEEDING UP COMPUTATIONAL PHYSICS WITH NEURAL NETWORKS

In recent years, there has been incredible interest in applications of neural networks to computational physics problems. One of the main pursuits being to emulate the desired physics for less computation then the physics simulation. Examples of this range from simulating 3D high energy particle showers seen in Paganini et al. (2017) to solving the Schrdinger equation seen in Mills et al. (2017). Computational Fluid Dynamics has gotten the most attention in this regard because of its many uses in engineering as well as computer animation (Tompson et al., 2016) (Hennigh, 2017). The prior work that is most related to our own is Guo et al. (2016) where they train a neural network to predict the steady state fluid flow from an objects geometry. Our method builds on this idea and we use the same general approach for approximating the fluid flow but with an improved architecture.

### 3.1 AUTOMATED DESIGN OPTIMIZATION OF AIRFOILS

To date, there has been substantial work in automated aerodynamic design for use in aeronautical and automotive applications (Ando et al., 2010) (Anderson & Aftosmis, 2015). Airfoil optimization in particular has received a lot of attention where the general methodology is to refine an airfoil geometry to minimize drag (Drela, 1998) (Koziel & Leifsson, 2013). Roughly speaking, there are two classes of optimization strategies used here. The first class being gradient free methods like simulated annealing, genetic algorithms, and particle swarm methods. A look at these methods and there applications to airfoil optimization can be found in Mukesh et al. (2012). The other class being gradient based methods such as steepest descent. Typically gradient based methods can perform optimization in fewer steps then gradient free methods however computing the gradient is often very costly. The simplest approach in doing so is finite difference method however this requires simulating the system a proportional number of times to the dimension of the search space in order to approximate the gradient. This is infeasible if the fluid simulation is computationally expensive and the search space is large. Our approach can be viewed as a gradient based method but where the gradients are coming from a neural network that is emulating the simulation.

In order to perform automated design of airfoils one needs to parameterize the space of possible geometries. There are a variety of approaches in doing this and a thorough list can be found in Salunke et al. (2014). In this work we use the parameterization technique found in Lane & Marshall (2009) and Hilton (2007) where the upper and lower surface are described by a polynomial and the parameters are the coefficients of this polynomial.

## 4 GRADIENT DECENT ON PARAMETER SPACE

An automated design optimization problem can be viewed in concrete terms as maximizing some desired fitness function $F(x)$, where $F : X \to \mathbb{R}$ for some space $X$ of design parameters.

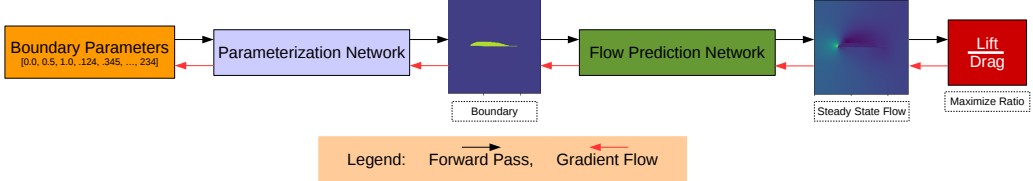

Figure 1: Illustration of Proposed Gradient Descent Method

$$\max_{\forall x \in X} F(x) \qquad (1)$$

In most real world setting, evaluating the fitness function $F$ can be computationally demanding as is the case with our fluid simulations. The first aspect of our method is to replace $F$ with a computationally efficient neural network $F_{net}$. This can offer considerable speed improvements as we will discuss bellow. The second piece of our method is the observation that $F_{net}$ is differentiable and can be used to obtain a usable gradient in the direction of maximizing fitness. This is in contrast to $F$ where it may be computationally infeasible to calculate the gradient and thus require other search techniques such as simulated annealing or genetic algorithms. Using this gradient allows faster optimization to be performed with fewer iterations as we will demonstrate bellow. There are some details that need to be addressed and to do so we go through the example problem of optimizing the fin heights on a heat sink.

In our heat sink problem, $X$ contains 15 real valued parameters between 0 and 1. Each of these parameters correspond to the height of an aluminum fin on the heat sink as seen in the figure 5.4. We also fix the amount of aluminum and scale the total length of all the fins to meet this requirement. This presents an interesting problem of determining the optimal length each fin should have to maximize the cooling of the heat source. The simplest application of our method is to use the 15 fin heights as inputs to a neural network that outputs a single value corresponding to the temperature at the heat source. This approach has the draw back that if you want to add another constraint to the optimization like making the left side cooler then the right side you would need to retrain the network. A solution to this problem is to have the network again take in the fin parameters but output the full heat distribution of the heat sink. This allows different quantities to be optimized but is still limiting in that our network only runs on a single parameter setup. Our solution to this problem is to train two networks. The first network, $P_{net}^{heat}$, takes in the fin parameters and generates a binary image corresponding to the geometry of the heat sink. We refer to this as the parameterization network. The second network, $S_{net}^{heat}$, predicts the steady state heat distribution from the geometry. Because the parameterization network is performing an extremely simple task and training data can be generating cheaply, we can quickly retrain $P_{net}^{heat}$ if we want to change the parameter space. The network $S_{net}^{heat}$ is now learning the more general task of predicting steady state heat flow on an arbitrary geometry. The same approach is used for the steady state flow problem and an illustration depicting this can be found in figure 4 . This approach allows our network to be as versatile as possible while still allowing it to used on many design optimization tasks.

Up until now we have not discussed how to generate the data needed to train these neural networks. Generating the data to train the parameterization network is relatively simple. If the parameterization is known, we simply make a set of parameter vectors and their corresponding geometries. In the case of the heat sink, this is a set of examples composed of the 15 parameters and there corresponding binary representation of the heat sink. Putting together a dataset for $S_{net}^{heat}$ or $S_{net}^{flow}$ (fluid flow network) is somewhat more complex. The simplest approach and the one used in this work is to simulate the respective physics on objects drawn from the object design space. For the heat sink problem this would entail a dataset of object geometries and their corresponding steady state heat distributions. This method has the disadvantage that the network only sees examples from the current

parameter search space and if it is changed the network may not be able to accurately predict the physics. We argue this is not a significant issue for two reasons. First, neural networks are very good at generalizing to examples outside their train set. An example of this can be seen in Guo et al. (2016) where the network is able to produce accurate fluid predictions on vehicle cross sections even though it was only trained on simple polygons. Second, it is easy to imagine a hybrid system where a network is trained on a large set of diverse simulations and then fine tuned on the current desired parameter space. For these reasons we feel that this approach of generating simulation data is not significantly limiting and does not detract from the generalizability of the approach.

### 4.1 FLOW PREDICTION NETWORK

In order for our method to work effectively we need a network to predict the pressure and velocity field of the steady state flow from an objects geometry. This is a difficult task because each point of flow is dependent on the entirety of the geometry. This global information requirement is met in the previous work (Guo et al., 2016) with a fully connected layer. This has drawbacks because fully connected layers are often slow, difficult to train, and parameter heavy. Our improved method keeps the entire network convolutional and employs a U-network architecture seen in Ronneberger et al. (2015) with gated residual blocks seen in Salimans et al. (2017). By making the network deep and using many downsamples and upsamples we can provide global information about the boundary when predicting each point of flow. Keeping the network all convolutional also allows the spacial information to be preserved. We found that the U-network style allowed us to train our network on considerably smaller datasets then reported in the previous work. The use of gated residual blocks also sped up training considerably. For input into the network we use a simple binary representation of the geometry instead of the Signed Distance Function representation used in the previous work as we found no benefit in this added complexity. The steady state heat prediction network uses the same basic network architecture and a complete description of all networks including the parametrization networks can be found in the appendix in figure 7.

## 5 EXPERIMENTS

In the following sections we subject our method and model to a variety of tests in order to see its performance.

### 5.1 DATASETS

To train the parameterization networks we generate a set of 10,000 examples for each system consisting of a parameter vector and their corresponding geometry. An example of what a heat sink geometry looks like can be found in figure 5.4. We use the parameterization found in Lane & Marshall (2009) for the 2D and 3D airfoils with 46 parameters that correspond to coefficients of a polynomial describing the upper and lower surface of the foil. A complete description of the parameterization can be found in the appendix.

The simulation datasets consists of 5,000, 5,000, and 2,500 training examples for the heat sink simulation, 2D fluid simulation, and 3D fluid simulation respectively. We use a 80-20 split in making the train and test sets. The geometries used for the simulations are drawn from the distributions used in the parameterization dataset. The heat simulations used a finite difference solver and the fluid flow simulation used the Lattice Boltzmann method.

### 5.2 TRAINING

We used the Adam optimizer for all networks (Kingma & Ba, 2014). For $S_{net}^{heat}$ and $S_{net}^{flow}$ a learning rate of 1e-4 was used until the loss plateaued and then the learning rate was dropped to 1e-5. Mean Squared Error was used as the loss function however when training the flow prediction network we scaled up the loss from the pressure field by a factor of 10 to roughly match the magnitude of the velocity vector field. The parameterization networks also used Mean Squared Error with a constant learning rate of 1e-4. We found the parameterization networks trained extremely quickly.

### 5.3 Gradient Decent Design Optimization Details

There are some complexities in how exactly the design parameters are optimized that need explanation. The most naive approach is to scale the computed gradient by some learning rate and add it to the design parameters. We found this approach effective however it was prone to finding local optimum. We found that adding momentum to the gradient reduced the chance of this and in most cases accelerated optimization. We also found that adding a small amount of noise too the parameters when computing gradients helped jump out of local optima. We used momentum 0.9 and a learning rate of 0.05 and 0.001 for the heat sink and airfoil problems respectively. The noise added to the parameters used a Gaussian distribution with mean 0 and standard deviation 0.01.

If the above approach is used naively it can result in parameter values outside of the original design space. To solve this problem we scale the input to the parameterization network between 0 and 1 and use a hard sigmoid to enforce this. This does not fix the problem completely though because if the parameters being trained leave the range of -0.5 to 0.5, the gradient will be zero and the parameter will be stuck at its current value. To prevent this we simply add a small loss that pushes any parameters outside the -0.5 to 0.5 range back in.

### 5.4 Heat Sink Optimization

As discussed above, the heat sink optimization task is to find a set of fin heights that maximally cool a constant heat source given a fixed total length of the fins. The set up roughly corresponds to an aluminum heat sink placed on a CPU where the heat source is treated as a continual addition of temperature. There is no heat dissipation between the underside of the heat sink but all other areas not on the heat sink are kept at a constant temperature. The intuitive solution to this optimization problem is to place long fins near the heat source and shorter fins farther away. Balancing this is a difficult task though because changing the length of any fin has a global effect on how much heat is dissipated by all the other fins.

After training our networks $P_{net}^{heat}$ and $S_{net}^{heat}$ we perform our proposed gradient optimization on the 15 fin heights to minimize the temperature at the source. In figure 5.4 we see the optimized heat sink and observe that the design resembles what our intuition tells us. We also note the extremely smooth optimization that occurs with only small bumps caused by the addition of noise noted above. A natural question to ask is how this compares to other search techniques. In order to answer these questions we use simulated annealing to search designs and use the original heat diffusion solver to evaluate their performance. In figure 5.4, we see that the optimized heat sink design produced by the neural network closely resembles that produced by simulated annealing. There are some minute differences however the total effectiveness in cooling the system are almost identical. We also note the iteration difference between the two methods. The gradient decent approach required roughly 150 iterations to converge where as the simulated annealing approach needed at least 800.

### 5.5 Flow Prediction Accuracy

Before we move to our final test of designing 2D and 3D airfoils it is important to know how accurately our model can predict steady state fluid flow. We can also verify our claim of a superior network architecture over previous work and show results indicating this. We omitted this discussion of accuracy from the heat sink problem however a figure showing the accuracy in predicting the heat at source can be found in figure 8 in the appendix.

The quantities of most interest in our predictions are the forces on the object. These are the values being optimized so being able to predict them accurately is of crucial importance. The forces are calculated from the pressure field by doing a surface integral over the airfoil. This can be done with any neural network library in a differentiable way by using a 3 by 3 transpose convolution on the boundary to determine the surface normals of the object. Then multiplying this with the pressure field and summing to produce the total force. Viscus forces are left out from this calculation as they are relatively small for thin airfoils. In figure 3, we see that our model is very accurate in predicting the forces. When comparing our network to the previous model we see a clear increase in accuracy. We also visually inspect the flow and see that the predicted flow is very sharp and doesn't have any rough or blurring artifacts.

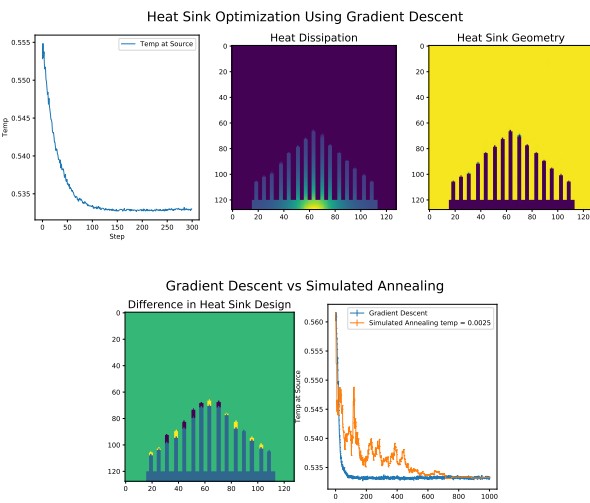

Figure 2: The top plot shows the optimization process and final design using our gradient descent method. The bottom plot shows a comparison of our optimization method to simulated annealing and the differences in final designs. As we can see, the gradient descent method converges much faster and finds roughly the same design.

## 5.6 Automated Design of 2D and 3D Airfoils

A conventional airfoil works by using a curved upper surface to create a low pressure zone and produce lift. The most important quantity for an airfoil is its lift drag ratio which in effect tells its efficiency. At different angles with respect to the fluid flow (angles of attack) the airfoil will produce different lift drag ratios. Roughly speaking, an airfoil should have a increase in lift drag ratio as the angle of attack increases until a max value is reached. For our optimization task, we maximize this lift drag ratio for an airfoil at angles of attack ranging from -5 to 17.5 degrees. The gradient for the airfoil is calculated 9 times at angles in this range and then combined to produce one gradient update. This approach of multiple angle optimization is common and can be found in Drela (1998). In figure 4 and 5 we see the optimized designs produced for the 2D and 3D simulations. We see that our method produces the expected shape and characteristic curve of lift drag ratio versus angle of attack. We also simulated the optimized airfoil with the Lattice Boltzmann solver and found that it performed similarly confirming that optimized designs produced by our method translate well to the original simulation.

We have seen that our method is quite effective at producing optimized designs but it is worth investigating what the fitness space looks like. To do this we selected a random airfoil and slowly changed one of its parameters to see the effect on the lift drag ratio. A plot of this can be seen in figure 5.6. We notice that while there are many local optima present, the change in lift drag ratio is very smooth and produces a very clean gradient. We view this as the reason our method optimizes so quickly. We found that local optima like the ones seen in this plot did not pose a serious problem during the optimization and when running multiple times with different starting designs the same basic shape was found with similar total fitness. We believe this was a result of both the momentum and addition of noise as well as optimizing multiple angles of attack at once. Adding this multiple angle constraint limits the number of possible designs and makes the chance of finding local optima smaller. We leave a deeper investigation into the effect of local optima for future work.

Similar to the heat sink problem, we compare our gradient decent method to simulated annealing. Unlike the heat sink problem though, performing simulated annealing with the Lattice Boltzmann solver was too computationally demanding and so we used our network to evaluate the lift drag ratio instead. We see from the figure 5.6 that using the gradient accelerates the optimization and in only

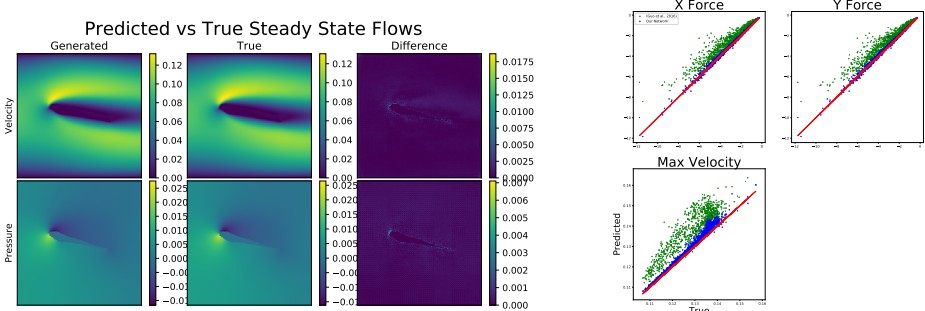

Figure 3: Comparison of steady state flow predicted by neural network and the Lattice Boltzmann flow solver. The right plot compares our network architecture (blue dots) with Guo et al. (2016) (green dots). As we can see, our network predicts forces and the max velocity more accurately then the other model.

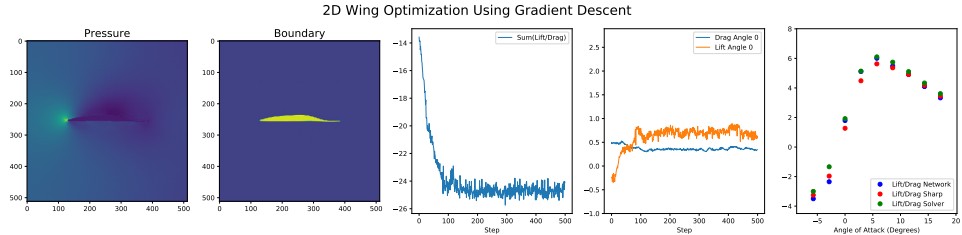

Figure 4: A 2D airfoil designed by our gradient descent method. The airfoil works by producing a low pressure area above its surface.

200 iterations it converges. In comparison, the simulated annealing requires at least 1500 iterations to reach similar performance.

## 5.7 COMPARISON OF COMPUTATION TIMES

The central purpose of our method is to accelerate the automated design process and in this section we attempt to quantify this in real time. The most important quantities are the time to perform a gradient update on the design parameters and the time needed to perform a simulation. Using these values we can give a very rough comparison of optimization using our method and other gradient free methods that use the flow solver. We leave this section for the airfoil design problems only.

The first quantity we look at is the raw speed of the fluid solver. We found that our flow solver converged to steady state in an average of 37.8 seconds for the 2D simulation and 163.8 seconds for the 3D simulation on a Nvidia 1080 GPU. We used the Sailfish library for these simulations as it performed faster then every other non-proprietary Lattice Boltzmann based fluid flow library (Januszewski & Kostur, 2014). In comparison to our neural network, performing one gradient update required only 0.052 seconds for the 2D simulation and 0.711 seconds for the 3D simulation. A more complete list of values including their relation to batch size can be found in the table 1 in the appendix. Given that performing automated design on the 2D airfoil required roughly 1,500 iterations at 9 different angles, this represents a total computation time of 141 hours. In comparison, our method only took 1.5 minutes to perform its 200 iterations at the 9 angles of attack. While this does represent a significant 5,000 times speed increase, we note that there are several methods of accelerating Lattice Boltzmann steady state flow calculations not explored in this work that under restricted conditions can give a significant speed increase (Guo & Shu, 2013) (Bernaschi et al., 2002). We also note that there are other applicable search methods such as genetic algorithms and particle swarm methods that may be more sample efficient. With the understanding that this

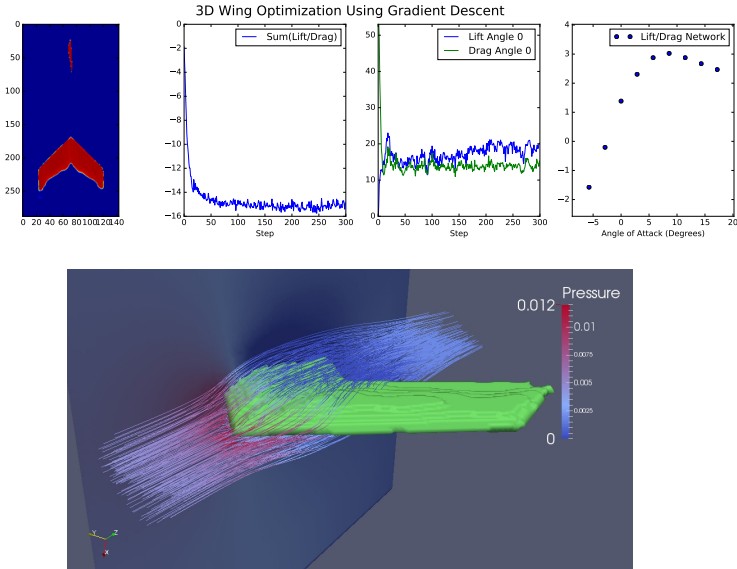

Figure 5: A 3D airfoil designed by our gradient descent method. The cross section of the wing is very similar to that seen in the 2D airfoil.

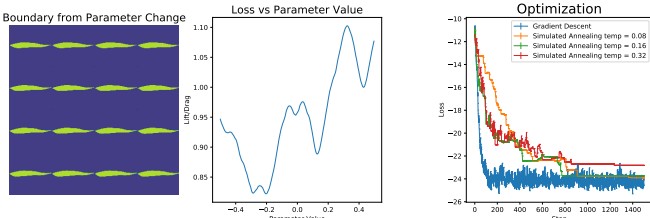

Figure 6: The left plot shows the change in lift drag ratio versus a change in a single design parameter. We note that while there are many local optima, the surface is very smooth and produces clean gradients. The right plot shows a comparison of the gradient decent optimization to simulated annealing in the 2D airfoil problem for a variety of starting temperatures.

comparison is somewhat rough, we view this result as strong evidence that our novel method is able to overcome some of the current computational limitations faced in automated design.

## 6 CONCLUSION

In this work we have presented a novel method for automated design and shown its effectiveness on a variety of tasks. Our method makes use of neural networks and gradient descent to provide powerful and fast optimization. There are many directions for future work such as applying this method to new domains like structural optimization and problems related to electromagnetism. One area of particular interest is design optimization on airfoils in turbulent time dependent flows. Another interesting area to explore is hybrid approaches where the neural network method is used to generate a rough design and then fine tuned with a high fidelity simulation.

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

# 7 APPENDIX

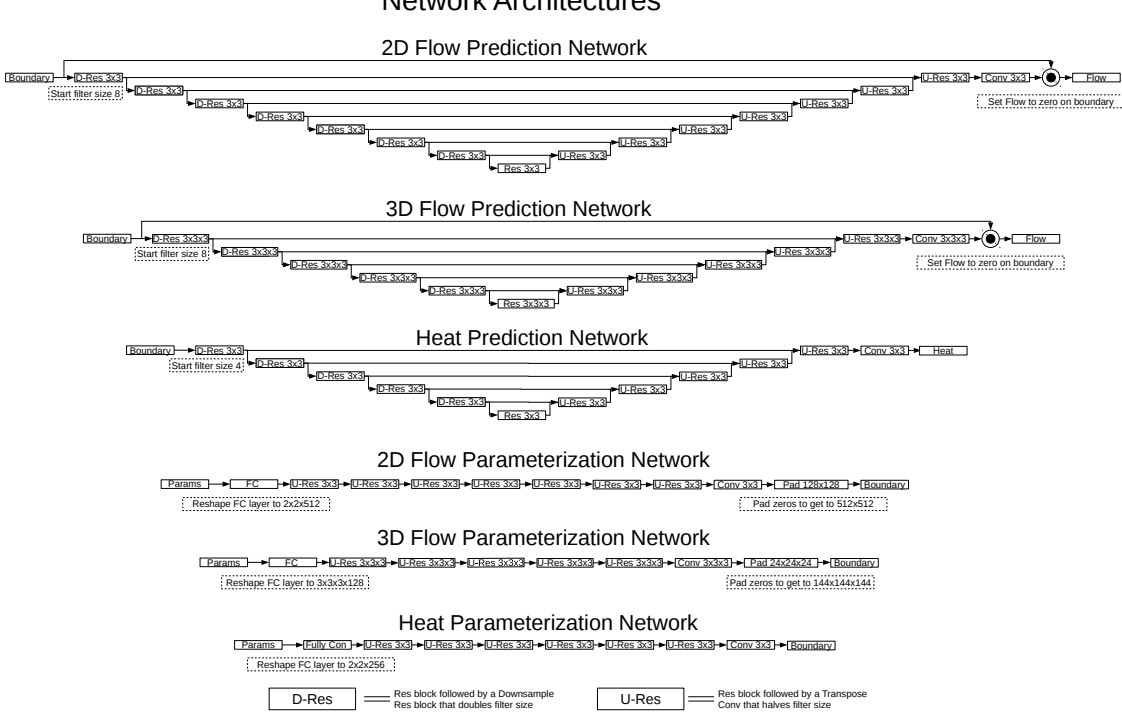

Figure 7: Complete network architectures.

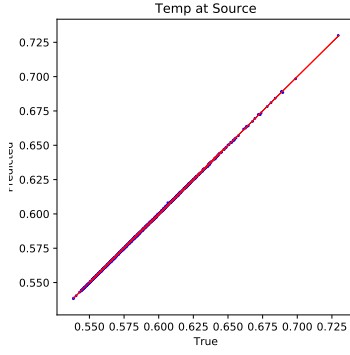

Figure 8: Difference between predicted heat at source of heat sink for our network and the original solver.

Table 1: Complete list of computation times for different network evaluations with different batch sizes.

| Batch Size | 1 | 2 | 4 | 8 | 16 |
|---|---|---|---|---|---|
| Flow Net $512^2$ | 0.020 sec | 0.017 sec | 0.016 sec | 0.015 sec | 0.015 sec |
| Param Net $512^2$ | 0.004 sec | 0.002 sec | 0.002 sec | 0.002 sec | 0.001 sec |
| Learn Step $512^2$ | 0.080 sec | 0.065 sec | 0.057 sec | 0.053 sec | 0.052 sec |
| Flow Net $144^3$ | 0.192 sec | 0.192 sec | 0.193 sec | 0.197 sec | Nan |
| Param Net $144^3$ | 0.025 sec | 0.024 sec | 0.025 sec | 0.025 sec | 0.025 sec |
| Learn Step $144^3$ | 0.911 sec | 0.711 sec | Nan | Nan | Nan |

## 7.1 AIRFOIL PARAMETERIZATION

The equation that parameterizes the upper and lower surface of the 2D airfoil is

$$S(x) = x^{n_1}(1-x)^{n_2} \sum_{i=0}^{N} A_i \frac{N}{i!(N-i)!} x^i (1-x)^{N-1} + hx \tag{2}$$

The parameters present are $n_1$, $n_2$, $A_i$s, and $h$. We also add the parameter $\theta$ that determines the angle of attack. In this work we fixed $n_1$ to 0.5 and $n_2$ to 1.0 as this will produce a rounded head on the airfoil. We also fix $h$ to zero making the tail the same height as the head. Thus the trainable parameters are the 42 values corresponding to the $A_i$s for the upper and lower surface. A illustration showing the parameterization can be found in figure 9. The 3D airfoil has similar parameterization.

$$S(\phi, y) = \phi^{n_1}(1-\phi)^{n_2} \sum_{i=0}^{N_x} \sum_{j=0}^{N_y} A_i B_j \frac{N_x}{i!(N_x-i)!} \phi^i (1-\phi)^{N_x-1} \frac{N_y}{i!(N_y-i)!} y^i (1-y)^{N_y-1} + h\phi \tag{3}$$

Where

$$\phi = \frac{x - sy}{(l - 0.5)y + 0.5} \tag{4}$$

This tells the height of the airfoil at a point $(x, y)$. The trainable parameters here are $n_1$, $n_2$, $A_i$s, $B_j$s, $h$, $s$, and $l$. Again, $n_1$, $n_2$, and $h$ are fixed to the values in the 2D case. We also have 2 parameters for the angle $\theta$ and $\psi$ that determine the rotation in the x and y direction. We keep $\psi$ at zero and only vary $\theta$ at the desired angles during the optimization. The parameters $s$ and $l$ correspond to the sweep present in the wing. This leaves the $A_i$s and $B_j$s for optimization. We split the remaining 39 parameters equally so that 13 values are used for $B_i$s and the remaining 26 are split between the $A_i$s for the upper and lower surface. For a much more in depth look at this parameterization, see Lane & Marshall (2009).

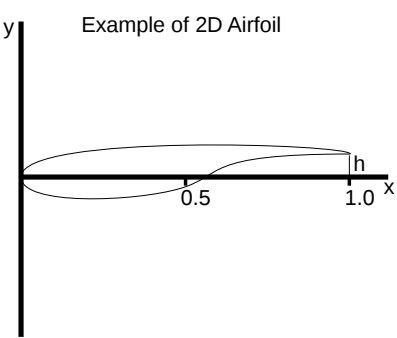

Figure 9: Example of a 2D airfoil.

