# OpenReview forum: "AUTOMATED DESIGN USING NEURAL NETWORKS AND GRADIENT DESCENT"
_ICLR.cc/2018/Conference — Invite to Workshop Track_

### Official Review · AnonReviewer2 · 2017-11-25
**Useful research direction, good results, not convinced about the generality of the prediction network**

**Rating:** 5
**Confidence:** 4

**Review:**

This paper proposes to use neural network and gradient descent to automatically design for engineering tasks. It uses two networks, parameterization network and prediction network to model the mapping from design parameters to fitness. It uses back propagation (gradient descent) to improve the design. The method is evaluated on heat sink design and airfoil design.

This paper targets at a potentially very useful application of neural networks that can have real world impacts. However, I have three main concerns:
1) Presentation. The organization of the paper could be improved. It mixes the method, the heat sink example and the airfoil example throughout the entire paper. Sometimes I am very confused about what is being described. My suggestion would be to completely separate these three parts: present a general method first, then use heat sink as the first experiment and airfoil as the second experiment. This organization would make the writing much clearer.

2) In the paragraph above Section 4.1, the paper made two arguments. I might be wrong, but I do not agree with either of them in general. First of all, "neural networks are good at generalizing to examples outside their train set". This depends entirely on whether the sample distribution of training and testing are similar and whether you have enough training examples that cover important sample space. This is especially critical if a deep neural network is used since overfitting is a real issue. Second, "it is easy to imagine a hybrid system where a network is trained on a simulation and fine tuned ...". Implementing such a hybrid system is nontrivial due to the reality gap. There is an entire research field about closing the reality gap and transfer learning. So I am not convinced by these two arguments made by this paper. They might be true for a narrow field of application. But in general, I think they are not quite correct.

3) The key of this paper is to approximate the dynamics using neural network (which is a continuous mapping) and take advantage of its gradient computation. However, many of dynamic systems are inherently discontinuous (collision/contact dynamics) or chaotic (turbulent flow). In those scenarios, the proposed method might not work well and we may have to resort to the gradient free methods. It seems that the proposed method works well for heat sink problem and the steady flow around airfoil, both of which do not fall into the more complex physics regime. It would be great that the paper could be more explicit about its limitations.

In summary, I like the idea, the application and the result of this paper. The writing could be improved. But more importantly, I think that the proposed method has its limitation about what kind of physical systems it can model. These limitation should be discussed more explicitly and more thoroughly.

---

### Official Review · AnonReviewer1 · 2017-11-26
**Interesting application that the ICLR community should learn about.**

**Rating:** 7
**Confidence:** 4

**Review:**

This paper introduces an appealing application of deep learning: use a deep network to approximate the behavior of a complex physical system, and then design optimal devices (eg airfoil shapes) by optimizing this network with respect to its inputs. Overall, this research direction seems fruitful, both in terms of different applications and in terms of extra machine learning that could be done to improve performance, such as ensuring that the optimization doesn't leave the manifold of reasonable designs.

 On one hand, I would suggest that this work would be better placed in an engineering venue focused on fluid dynamics. On the other hand, I think the ICLR community would benefit from about the opportunities to work on problems of this nature.

 =Quality=
The authors seem to be experts in their field. They could have done a better job explaining the quality of their final results, though. It is unclear if they are comparing to strong baselines.

=Clarity=
The overall setup and motivation is clear.

=Originality=
This is an interesting problem that will be novel to most member of the ICLR community. I think that this general approach deserves further attention from the community.


=Major Comments=
* It's hard for me to understand if the performance of your method is actually good. You show that it outperforms simulated annealing. Is this the state of the art? How would an experienced engineer perform if he or she just sat down and drew the shape of an airfoil, without relying on any computational simulation at all?

* You can afford to spend lots of time interacting with the deep network in order to optimize it really well with respect to the inputs. Why not do lots of random initializations for the optimization? Isn't that a good way to help avoid local optima?

* I'd like to see more analysis of the reliability of your deep-network-based approximation to the physics simulator. For example, you could evaluate the deep-net-predicted drag ratio vs. the simulator-predicted drag ratio at the value of the parameters corresponding to the final optimized airfoil shape. If there's a gap, it suggests that your NN approximation might have not been that accurate.

=Minor Comments=
* "We also found that adding a small amount of noise too the parameters when computing gradients helped jump out of local optima"
Generally, people add noise to the gradients, not the values of the parameters. See, for example, uses of Langevin dynamics as a non-convex optimization method.

* You have a complicated method for constraining the parameters to be in [-0.5,0.5]. Why not just enforce this constraint by doing projected gradient descent? For the constraint structure you have, projection is trivial (just clip the values).

 * "The gradient decent approach required roughly 150 iterations to converge where as the simulated annealing approach needed at least 800."
This is of course confounded by the necessary cost to construct the training set, which is necessary for the gradient descent approach. I'd point out that this construction can be done in parallel, so it's less of a computational burden.

* I'd like to hear more about the effects of different parametrizations of the airfoil surface. You optimize the coefficients of a polynomial. Did you try anything else?

* Fig 6: What does 'clean gradients' mean? Can you make this more precise?

* The caption for Fig 5 should explain what each of the sub figures is.

---

### Official Review · AnonReviewer3 · 2017-11-26
**This is a good deep learning application paper without sufficient algorithmic/theoretical novelty suitable for ICLR.**

**Rating:** 4
**Confidence:** 5

**Review:**

1. This is a good application paper, can be quite interesting in a workshop related to Deep Learning applications to physical sciences and engineering
2. Lacks in sufficient machine learning related novelty required to be relevant in the main conference
3. Design, solving inverse problem using Deep Learning are not quite novel, see
Stoecklein et al. Deep Learning for Flow Sculpting: Insights into Efficient Learning using Scientific Simulation Data. Scientific Reports 7, Article number: 46368 (2017).
4. However, this paper introduces two different types of networks for "parametrization" and "physical behavior" mapping, which is interesting, can be very useful as surrogate models for CFD simulations
5. It will be interesting to see the impacts of physics based knowledge on choice of network architecture, hyper-parameters and other training considerations
6. Just claiming the generalization capability of deep networks is not enough, need to show how much the model can interpolate or extrapolate? what are the effects of regulariazations in this regard?

---

### Decision · Program_Chairs · 2018-01-29
**ICLR 2018 Conference Acceptance Decision**

**Decision:**

Invite to Workshop Track

**Comment:**

Differentiable neural networks used as a measure of design optimality in order to improve efficiency of automated design.


Pros:
- Genetic algorithms, which are the dominant optimization routine for automated design systems, can be computationally expensive. This approach alleviates this bottleneck under certain circumstances and applications.

Cons:
- Primarily application paper, machine learning advancement is marginal.
- Multiple reviewers: Generalization capability not clear. For example, some utility systems may be stochastic (i.e. turbulence) and require multiple trials to measure fitness, which this method would not be able to model.
Overall, the committee feels this paper is interesting enough to appears as a workshop paper.